# Natural oxidase-mimicking copper-organic frameworks for targeted identification of ascorbate in sensitive sweat sensing

Zhengyun Wang[1], Yuchen Huang[2], Kunqi Xu[3], Yanyu Zhong[1], Chaohui He[1], Lipei Jiang[1], Jiankang Sun[1], Zhuang Rao[1], Jiannan Zhu[1], Jing Huang[1], Fei Xiao[1], Hongfang Liu [1] ✉ & Bao Yu Xia [1] ✉

Sweat sensors play a significant role in personalized healthcare by dynamically monitoring biochemical markers to detect individual physiological status. The specific response to the target biomolecules usually depends on natural oxidase, but it is susceptible to external interference. In this work, we report tryptophan- and histidine-treated copper metal-organic frameworks (Cu-MOFs). This amino-functionalized copper-organic framework shows highly selective activity for ascorbate oxidation and can serve as an efficient ascorbate oxidase-mimicking material in sensitive sweat sensors. Experiments and calculation results elucidate that the introduced tryptophan/histidine fundamentally regulates the adsorption behaviors of biomolecules, enabling ascorbate to be selectively captured from complex sweat and further efficiently electrooxidized. This work provides not only a paradigm for specifically sweat sensing but also a significant understanding of natural oxidase-inspired MOF nanoenzymes for sensing technologies and beyond.

Wearable sensors can continuously provide health-related parameters by tracking the biological fluid of the human body[1–4]. Sweat rich in metabolites and electrolytes can be monitored electrochemically in real time to reflect an individual's physiological status at the molecular level, which is essential for personalized healthcare[5–9]. However, owing to the high complexity of sweat components, it is difficult to achieve a stable and selective response to target sweat biomarkers, especially biomolecules[10]. Basically, the responses to sweat biomolecules, such as ascorbate, lactate, and uric acid, are determined by their electro-oxidation behavior. These biomolecules have similar oxidation potentials, and their corresponding oxidation reactions occur simultaneously and cannot be effectively distinguished[11]. As a bioreceptor, natural oxidase can selectively catalyze substrate biomolecules and produce specific responses[12–17]. However, depending on the active characteristics of dissolved oxygen and its rapid deactivation in vitro, the performance of the sensor is rapidly decreased[18], which seriously limits the practical application of oxidase in sweat sensors.

In recent years, the development of nanoenzymes has brought opportunities for sweat biomolecular sensing[19,20]. Some nanoenzymes, such as nanoscale carbon and metal-based materials, show high activity against sweat biomolecules and are regarded as natural oxidase mimetics[21–23]. However, compared with natural oxidases, common nanoenzymes do not contain any recognition sites to specifically lock biomolecules[24], and sweat-containing complex biomolecules will be adsorbed and electrooxidized indiscriminately. Consequently, these open-ended molecular oxidation behaviors and low substrate

[1]Hubei Key Laboratory of Material Chemistry and Service Failure, Key Laboratory of Material Chemistry for Energy Conversion and Storage (Ministry of Education), Hubei Engineering Research Center for Biomaterials and Medical Protective Materials, School of Chemistry and Chemical Engineering, Huazhong University of Science and Technology, 1037 Luoyu Rd, 430074 Wuhan, PR China. [2]Secretariat license de chimie, bâtiment 460, Université Paris-saclay, 91400 Orsay Paris, France. [3]Key Laboratory of Inorganic Functional Materials and Devices, Shanghai Institute of Ceramics, Chinese Academy of Sciences, 201899 Shanghai, PR China. ✉e-mail: liuhf@hust.edu.cn; byxia@hust.edu.cn

selectivity cannot realize highly sensitive identification and the response of targeted biomolecules[25]. Therefore, it remains a great challenge to customize nanoenzymes and achieve specific capture and electrooxidation of targeted sweat biomolecules.

In this work, we report an effective histidine (His)- and tryptophan (Trp)-functionalized copper metal-organic frameworks (Cu-MOFs), $Cu(C_{10}O_5H_8)$ (HT-STAM-17-OEt)), which was successfully achieved by epitaxial growth and postsynthetic treatment with Trp and His. HT-STAM-17-OEt with a similar ascorbate oxidase structure can effectively function in sweat sensors, with a high sensitivity of 0.18 and 0.48 mA cm$^{-2}$ mM$^{-1}$ in acidic and alkaline sweat, respectively. Comprehensive experimental results and theoretical calculations reveal that HT-STAM-17-OEt follows a working mechanism similar to that of natural oxidase; that is, amino acids induce the specific capture and recognition of ascorbate and then rapidly oxidize it on the copper sites. This work demonstrates an effective MOF nanoenzyme platform inspired by

natural oxidase, which may provide significant understanding for targeted identification in sensing technologies and beyond.

## Results

### Structural characterization of ascorbate oxidase mimicking Cu-MOF

Among many sweat biomolecules, ascorbate is one of the most important biomarkers related to nutrition and immunity (Fig. 1a)[26]. Ascorbate plays an indispensable role in many body functions, such as iron absorption, collagen production, infection protection, and neurological disorder prevention[27–30]. Thus, ascorbate sensing in sweat is of significance in evaluating and preventing the risks of these diseases. Ascorbate is specifically captured and recognized by natural ascorbate oxidase (Fig. 1b), which has a pocket formed by Trp/His and copper sites as specific binding and catalytic centers for precise capture and rapid oxidation, respectively[31,32]. Essentially, the specific binding

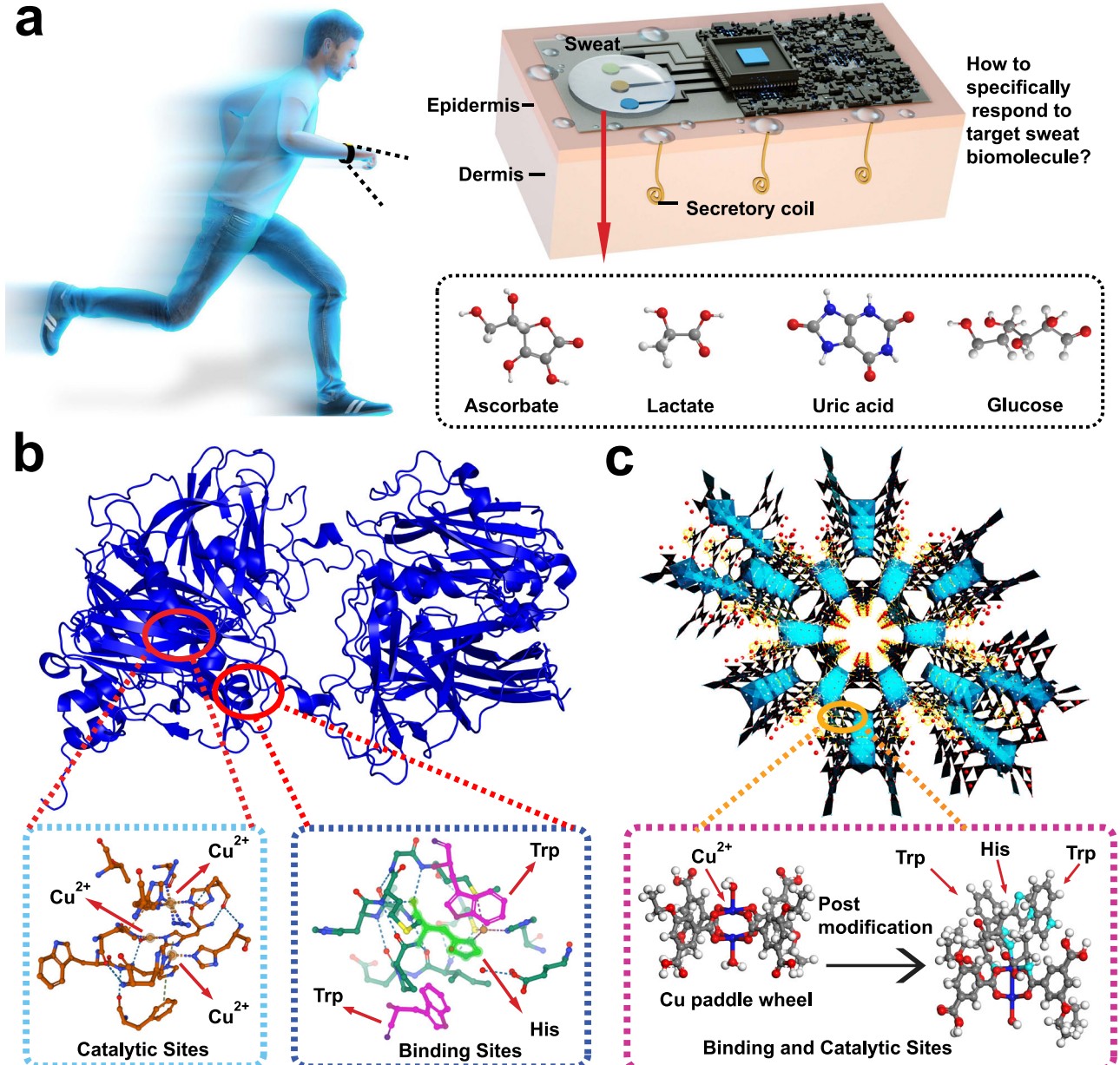

**Fig. 1 | Customizing nanoenzymes inspired by natural ascorbate oxidase for sweat ascorbate sensing. a** Sweat sensors for noninvasive health monitoring, **b** natural ascorbate oxidase with Cu catalytic and specific binding sites formed by

histidine (His) and tryptophan (Trp) for selective catalysis of ascorbate, and **c** concept of ascorbate oxidase mimicking MOF. The blue, red, gray, and white balls represent Cu, O, C, and H atoms, respectively.

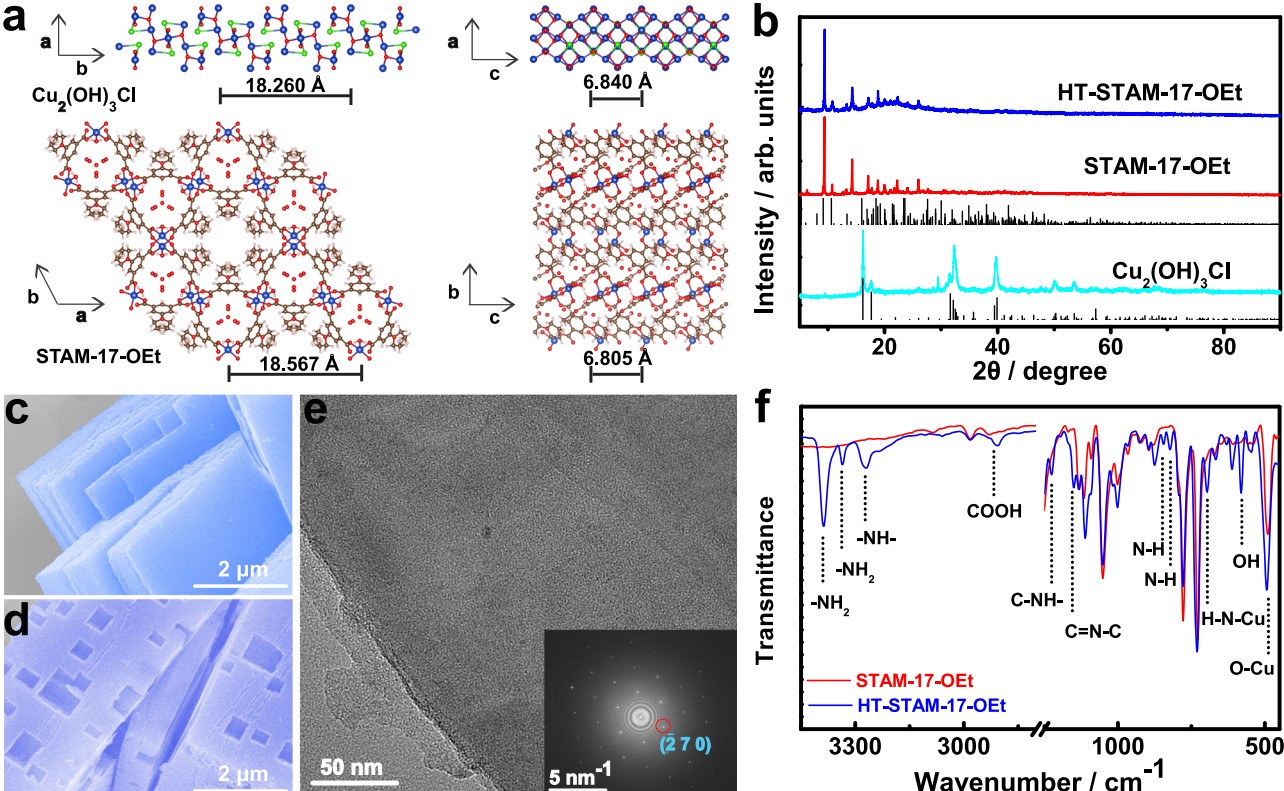

**Fig. 2 | Structure characterizations of STAM-17-OEt and HT-STAM-17-OEt.**
**a** Crystal structures of Cu₂(OH)₃Cl and STAM-17-OEt. The blue, red, brown, green, and gray balls represent Cu, O, C, Cl, and H atoms, respectively. **b** XRD patterns of Cu₂(OH)₃Cl, STAM-17-OEt and HT-STAM-17-OEt. FESEM images of **c** STAM-17-OEt and **d** HT-STAM-17-OEt. **e** HRTEM image of HT-STAM-17-OEt; the inset shows the selected area electron diffraction pattern. **f** ATR-IR spectra of STAM-17-OEt and HT-STAM-17-OEt.

behavior of ascorbate oxidase originates from the high affinity of Trp/His to ascorbate, and its oxidation capability depends on the redox of copper species. Although some nanoenzyme-related research works have demonstrated that CuO nanoparticles[33] and Pt nanoparticles[34,35] both have ascorbate oxidase-like activity and can efficiently catalyze ascorbate, these artificial nanoenzymes cannot specifically recognize ascorbate owing to chemical structure limitations. Based on these results, STAM-17-OEt with excellent hydrolytic stability is selected to mimic natural ascorbate oxidase because STAM-17-OEt with kagome-type topology centered by Cu paddle wheel clusters may show the electro-oxidation behavior of ascorbate under an electrical field[36]. The initial STAM-17-OEt does not contain selective binding sites for ascorbate. We introduced Trp and His on Cu paddlewheel clusters by postsynthetic amino acid treatment, trying to change the capture behavior of MOF to ascorbate and make it specific (Fig. 1c).

Cu₂(OH)₃Cl is selected to induce the formation of STAM-17-OEt in light of their close crystal axes (Fig. 2a). The X-ray diffraction (XRD) patterns in Fig. 2b show that STAM-17-OEt can be obtained by using Cu₂(OH)₃Cl nanocuboids as the copper source. No distinct XRD peak variation is observed in HT-STAM-17-OEt after postsynthetic treatment by mixing His and Trp at moderate concentrations, which indicates that the whole crystal structures of STAM-17-OEt are not obviously destroyed after amino acid soaking and ligand exchanging reaction does not occur in this case. Field-emission scanning electron microscopy (FESEM) images of STAM-17-OEt in Fig. 2c show the microslate structure with a smooth surface and layered stack (Supplementary Fig 5). However, some small cubic holes can be clearly observed on the surface (Fig. 2d), and the microslate architectures remain. The pore shape depends on the concentration of mixed amino acids (Supplementary Fig. 7). Moreover, smooth and flat internal exfoliation layers in HT-STAM-17-OEt are observed by high-resolution transmission

electron microscopy (HRTEM) (Fig. 2e). The selected-area electron diffraction (SAED) pattern of the inner exfoliation layers shows the ordered lattice of (2̄70) (Fig. 2e inset), indicating the single crystal structure[37–39]. The unit cell parameters of Cu₂(OH)₃Cl are 6.010, 9.130, and 6.840 Å for *a*, *b*, and *c*, respectively[40]. STAM-17-OEt closely matches the lattice parameters with *a* (18.567 Å) and *c* (6.805 Å). Therefore, the growth of the *ac* plane of STAM-17-OEt can align with the *bc* plane of Cu₂(OH)₃Cl, i.e., heteroepitaxial growth. In addition, continuous ionization of 5-ethoxyisophthalic acid induces the slow dissolution of Cu₂(OH)₃Cl, which leads to a gradual size increase for the STAM-17-OEt crystal. Therefore, the STAM-17-OEt crystal inherits the cuboid morphology of Cu₂(OH)₃Cl and finally forms a slat-like structure. On the other hand, since the defects formed on crystals come from the etching of Lewis acids[41,42], the HRTEM image also indicates this etching of Trp/His only remains on the crystal surface. The integrity crystal structure of STAM-17-OEt indicates that the absorption of amino acids is dominant on the crystal surface and only a small amount of Trp or His may diffuse into the micropores. Elementary mapping results confirm the successful modification of amino acids on HT-STAM-17-OEt crystal surface (Supplementary Fig. 6).

Attenuated total reflectance infrared spectrum (ATR-IR) was further employed to examine STAM-17-OEt and HT-STAM-17-OEt. As shown in Fig. 2f, the peaks at 3387 and 3334 cm⁻¹ correspond to N–H scissoring, and the peak at 3269 cm⁻¹ corresponds to NH stretching vibrations. The peaks at 2907, 1226, 845, and 581 cm⁻¹ are attributed to O–H stretching (COOH), C–NH– stretching, N–H in-plane rocking, and O–H out-of-plane deformation in Trp, respectively. The peaks 1150 and 822 cm⁻¹ are assigned to C=N–C stretching and N–H in-plane rocking vibrations in His, respectively. Apart from the characteristic bands of His and Trp, HT-STAM-17-OEt shows a characteristic peak of N–Cu– (696 cm⁻¹), which is attributed to Cu–N stretching vibrations.

Additionally, Cu−O stretching increases from 490 to 494 cm$^{-1}$, which originates from the influence of the coordination of NH$_2$ to the Cu paddle wheel structure. The ATR-IR results demonstrate that His and Trp attack the Cu paddle wheel and coordinate to Cu on STAM-17-OEt. Therefore, the surficial Cu paddle wheel on STAM-17-OEt could be confirmed as the loading site for His and Trp. The ratio of Trp/His is calculated as 1.88 by normalization of the common peak (O−H out of plane bending) in His and Trp (Supplementary Fig. 8). The quantity loading amounts and ratios of amino acids are investigated by liquid chromatograph mass spectrometer (LC−MS). According to the established standard curve and dilution relationship (20 times), the loading amount of Trp and His on HT-STAM-17-OEt (100 mg) is calculated as $3.4776 \times 10^{-2}$ and $1.7796 \times 10^{-2}$ mmol (Supplementary Fig. 10), respectively. Thus, the mass loading of Trp and His is calculated as 7.10% and 2.76%, respectively. And the mole ratio of Trp/His can be counted as 1.95, which is near to the result of ATR-IR analysis (1.88:1). And the mole ratio of Trp/His can be counted as 1.95, which is consistent with ATR-IR result (Supplementary Fig. 8) and the ratio of amino acids (2:1) initially used for treatment. Moreover, X-ray photoelectron spectroscopy (XPS) demonstrates a negative shift of binding energy for Cu 2$p$ (0.17 eV), and electron paramagnetic resonance (EPR) shows an additional Cu$^{2+}$ resonance signal in HT-STAM-17-OEt (Supplementary Figs. 11 and 12), suggesting a surface Cu coordination structure different from that of STAM-17-OEt. These results also prove that His and Trp interact with copper through coordination.

## Electrochemical performances of HT-STAM-17-OEt nanoenzymes

Figure 3a shows identical cyclic voltammetry (CV) curves for STAM-17-OEt and HT-STAM-17-OEt in 0.1 M NaCl solution, which suggests that surface amino acids will not change the electrochemical behavior of STAM-17-OEt. The peaks at 0.3 and −0.2 V (vs. Ag/AgCl) correspond to the Cu(I)/Cu(II) redox processes, while the peak at 0.05/−0.55 V (vs. Ag/AgCl) is attributed to the irreversible redox of Cu(0)/Cu(I)[43]. However, both STAM-17-OEt and HT-STAM-17-OEt

exhibit a dramatically enhanced current response in the presence of ascorbate, manifesting the electrochemical activity of both MOFs toward ascorbate. A selectivity study was carried out by using a high concentration of interfering biomolecules in sweat (Fig. 3b). We find those interfering biomolecules, including uric acid, glucose, ethanol, and cortisol, with their corresponding highest concentrations in sweat, do not generate interference signals with STAM-17-OEt at 0.5 V (vs. Ag/AgCl), but the current response to ascorbate (10 µM) is directly offset by lactate (20 mM). In addition, further lowering the applied potential would increase the response of ascorbate as well as interference of lactate (Supplementary Fig. 15a). This phenomenon indicates that ascorbate and lactate are both easily adsorbed and oxidized by STAM-17-OEt and those other molecules, including glucose and uric acid, cannot. In comparison, HT-STAM-17-OEt has good selectivity in that any interference would not be caused by high concentrations of lactate, uric acid, glucose, ethanol, and cortisol, and the current response to ascorbate increases with increasing applied potential (Supplementary Fig. 15b). Different from high concentrations, these interfering biomolecules at low concentrations (50 nM) cannot generate any interference current signals for HT-STAM-17-OEt and STAM-17-OEt (Supplementary Fig. 16). Moreover, HT-STAM-17-OEt can quickly respond to ascorbate when ascorbate is added with high concentrations of interfering biomolecules at the same time, while STAM-17-OEt has no response in this case (Supplementary Fig. 17). This result demonstrates that HT-STAM-17-OEt can favorably recognize ascorbate from other biomolecules, but STAM-17-OEt cannot. To explore the action of His and Trp, we further use one type of amino acid to modify STAM-17-OEt to study the selectivity. It is found that STAM-17-OEt treated with His or Trp alone still had lactate interference (Supplementary Fig. 18). Integrating with ATR-IR results, these findings further illustrate that the structures composed of His and Trp on the STAM-17-OEt surface, rather than structures composed of the same amino acids, determine ascorbate specificity due to higher affinity to ascorbate. It should be noted that ascorbate specificity cannot be achieved by using Trp/His at

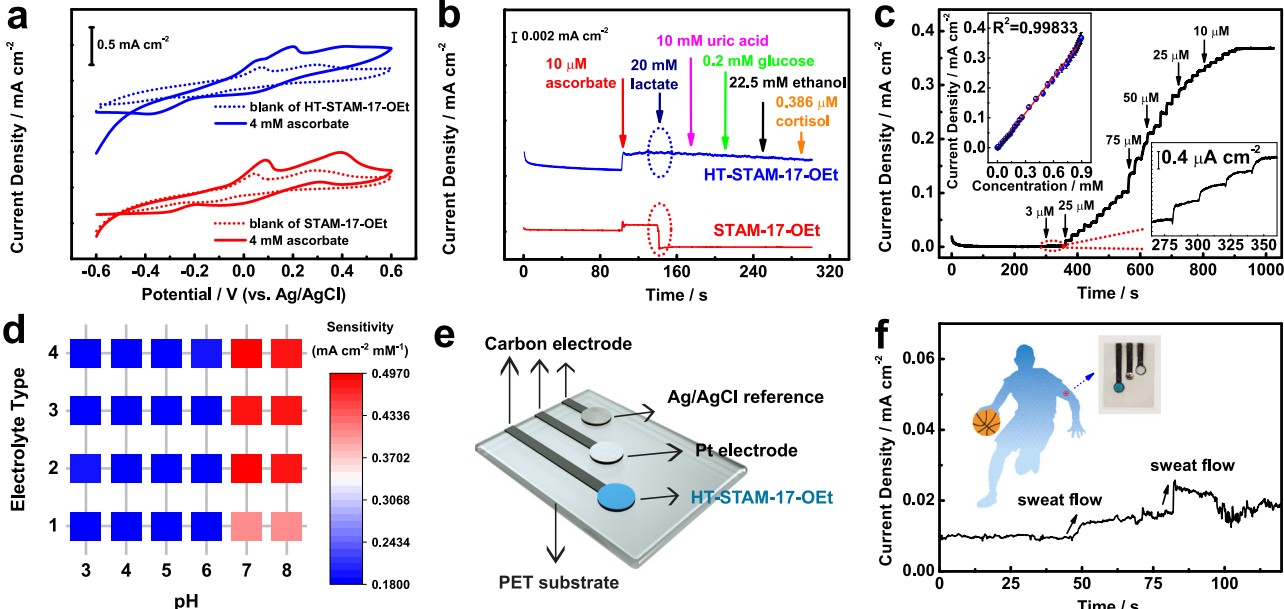

**Fig. 3 | Electrochemical behavior and ascorbate sensing performances of HT-STAM-17-OEt. a** CV curves and **b** amperometry response at 0.5 V vs. Ag/AgCl for STAM-17-OEt and HT-STAM-17-OEt in 0.1 M NaCl (pH = 7) at 20 mV s$^{-1}$. **c** Amperometry response of HT-STAM-17-OEt to successive addition of ascorbate at 0.5 V vs. Ag/AgCl and the corresponding calibration curve (relative standard deviation was obtained from three groups of samples). **d** Sensitivity map of HT-STAM-17-OEt in different electrolytes with different pH values, where the 1–4 electrolyte types represent aqueous solutions containing 0.1 M NaCl, 0.1 M NaCl/ 18.5 mM KCl, 0.1 M NaCl/18.5 mM KCl/12.4 mM CaCl$_2$, 0.1 M NaCl/18.5 mM KCl/ 12.4 mM CaCl$_2$/1 mM NH$_4$Cl, respectively. **e, f** Platform model supported by polyethylene glycol terephthalate (PET) substrate and data collection of the HT-STAM-17-OEt-based sensor.

relatively high (50 mM/25 mM) or low concentrations (0.5 mM/ 0.25 mM) to treat STAM-17-OEt (Supplementary Fig. 19).

The amperometry measurements are performed at 0.5 V (vs. Ag/ AgCl) to analyze the ascorbate sensing performance of HT-STAM-17-OEt. As shown in Fig. 3c, the current densities increase stepwise on aliquot additions of ascorbate, and well-defined steady-state current responses are achieved within 5 s. The HT-STAM-17-OEt-based ascorbate sensor shows a linear dynamic range over the concentration from 3 μM to 0.897 mM with a high sensitivity of 0.41 mA cm$^{-2}$ mM$^{-1}$ and a low detection limit of 1.0 μM (signal-to-noise ratio S/N = 3) in neutral NaCl solution. In addition, the HT-STAM-17-OEt-based ascorbate sensor exhibits good repeatability and stability (Supplementary Figs. 21, 22), much superior to the performance of ascorbate oxidase-based electrodes (Supplementary Fig. 23). Considering the actual sweat sensing application, we evaluated the sensitivity of the HT-STAM-17-OEt-based ascorbate sensor in different electrolytes in the presence of different ions (Fig. 3d). The sensitivity of the HT-STAM-17-Oet-based ascorbate sensor shows different levels in different types of electrolytes: 0.18, 0.41, and 0.48 mA cm$^{-2}$ mM$^{-1}$ for NaCl with other ions, neutral and alkaline NaCl, neutral and alkaline NaCl with other ions, respectively. This means that electrolyte pH influences the actual sweat ascorbate sensitivity of HT-STAM-17-OEt. The sensitivity discrepancy originates from the different charge transfer resistances of HT-STAM-17-OEt in different electrolytes (Supplementary Fig. 24). In addition, a mixture of high (20 mM) and low (50 nM) concentrated sweat biomolecules, including lactate, uric acid, glucose, ethanol, and cortisol, did not influence the response to ascorbate of HT-STAM-17-OEt in artificial salt solutions with different pH values (5–8) (Supplementary Fig. 25). These results demonstrate the suitability of HT-STAM-17-OEt as an oxidase-mimicking electrocatalyst for the determination of sweat ascorbate levels based on the relationship between sweat pH and sensitivity. Furthermore, we integrated HT-STAM-17-OEt, Ag/AgCl reference, and Pt electrodes on a polyethylene glycol terephthalate (PET) substrate to construct a sensor platform and explore the functionality in actual sweat ascorbate sensing (Fig. 3e). At an applied potential of 0.5 V (vs. Ag/AgCl), this sensing platform is highly sensitive to continuous sweat flow and gives a fast current response to sweat ascorbate with a low background current (0.01 mA cm$^{-2}$) within 5 s (Fig. 3f). By evaluating the pH value of sweat pH (pH = 5.0) and reading the current response, the sweat ascorbate concentration was measured to be 38.9 μM.

## Specific binding behavior analysis of HT-STAM-17-OEt nanoenzymes

We first studied the binding of HT-STAM-17-OEt to ascorbate and lactate by isothermal titration calorimetry (ITC). As shown in Fig. 4a, the ITC result reveals the spontaneous interaction between HT-STAM-17-OEt and ascorbate (Gibbs free energy, $\triangle G < 0$) and favored enthalpy ($\triangle H < 0$). Additionally, the number of ascorbate adsorption sites per Cu unit is 2.58 for HT-STAM-17-OEt. HT-STAM-17-OEt has a higher level of affinity ($K$) and a more negative enthalpy change ($\triangle H = -5.67 \times 10^4$ cal mol$^{-1}$) for ascorbate, which indicates a more favorable binding effect of Trp/His on ascorbate compared with STAM-17-OEt (Supplementary Fig. 26a). In comparison, low heat compensation for binding to lactate of HT-STAM-17-OEt can be observed, indicating that Trp/His on STAM-17-OEt has a much higher affinity for ascorbate than lactate. The ascorbate binding of HT-STAM-17-OEt exhibits a negative entropy change ($\triangle S = -171$ cal mol$^{-1}$ deg$^{-1}$), illustrating the dominant role of hydrogen bonding and van der Waals forces in the interaction of HT-STAM-17-OEt and ascorbate. Importantly, $\triangle G$, $\triangle H$, and $\triangle S$ with all negative values are generally considered signs of specific interactions for ligand binding of proteins[44,45]. Consequently, it is certain that Trp/His structures enable ascorbate capture to switch from open type to specificity. Through exploration of the influence on changing the Trp/His modification sequence, we

discovered that STAM-17-OEt was modified by first using Trp and then His still has lactate interference. STAM-17-OEt modified by first using His and then Trp also has lactate interference (Supplementary Fig. 27). This result hints that His/Trp structures that can preferentially bind to ascorbate would not form by changing the modification order owing to full occupation of Trp or His on the Cu paddle wheel. Based on ATR-IR results, −HN−Cu originates in two ways. One is the direct axial coordination of amino acids, and the other is that amino acids connect to ligands through destruction on the surface Cu paddle wheel due to Cu Lewis acid property-induced activation of carbonyl carbon. Thus, there are six possible structures of amino acids on the Cu paddle wheel (Supplementary Fig. 28). Due to the overall ratio of Trp/His (2:1) on the STAM-17-OEt surface, the structure in which one His is connected to Cu along the axial direction and two Trp are connected to Cu on both sides is the most reasonable structure because the central amino acid can be shared by two different amino acids, ensuring the combined effect of Trp/His. Such Trp/His on Cu paddle wheel structures dominate on the MOF surface and is responsible for ascorbate selectivity.

We then used density functional theory calculation (DFT) to explore the interaction with ascorbate and lactate of this Trp/His structure (Fig. 4b). After geometric optimization, one His and two Trp could form two pocket-like structures on the copper paddlewheel by coordination. For ascorbate binding, due to lactone rings and multiple hydroxyl groups in ascorbate, this Trp/His-formed pocket with appropriate geometry could effectively fix ascorbate through π–π stacking and more hydrogen bonds, forming a staggered system with the lactone ring of ascorbate and the side chain of Trp as well as His. For lactate binding, a single lactate molecule could also be adsorbed on the Trp/His pocket by a single hydrogen bond. However, in comparison, the energy for ascorbate adsorption is calculated as −0.78 eV, which is more negative than the lactate adsorption energy, suggesting that ascorbate capture on the Cu paddle wheel is a spontaneous process, whereas lactate capture requires additional energy. Calculation results show that ascorbate binding occurs more easily than lactate binding because of the more negative Gibbs free energy. This also reasonably explains the ascorbate selectivity phenomenon. Such Trp/His pockets on the STAM-17-OEt surface determine the significant specific capture of ascorbate, and other biomolecules would be directly excluded (Fig. 4c). Interestingly, ascorbate specificity of other Cu-MOFs with a Cu paddle wheel can be achieved by using this post-synthesis modification approach (Supplementary Fig. 29), suggesting the universality of Trp/His structure formation for specific binding to ascorbate.

## Electrooxidation mechanism analysis of HT-STAM-17-OEt nanoenzymes

Figure 5a presents the potentiodynamic polarization curves of HT-STAM-17-OEt in 0.1 M NaCl (pH = 7) in the absence and presence of ascorbate. The current density values for anodic reactions increase with increasing overpotential, indicating the diffusion-determined anodic process. This result indicates that HT-STAM-17-OEt would be activated to species with strong oxidation capability under positive potential. In the presence of 0.2 mM ascorbate, no clear change in the polarization curve was observed. In the presence of 2 mM ascorbate, the anode current densities increase and remain nearly constant across all overpotential values, demonstrating that diffusion-controlled ascorbate electrooxidation could further facilitate the electric field-induced activation of HT-STAM-17-OEt. In situ electrochemical infrared spectroscopy (in situ EC-IR) was employed to directly observe the spectroscopy signal variation of HT-STAM-17-OEt under an electric field (0.5 V vs. Ag/AgCl). As seen in Fig. 5b, the adsorption peak corresponding to the −OH stretching vibration was observed after applying a potential for 1 min, and the adsorption peak intensities slightly increased after 10 min. The in situ EC-IR results suggest that −OH groups are eliminated in HT-STAM-17-OEt under a positive

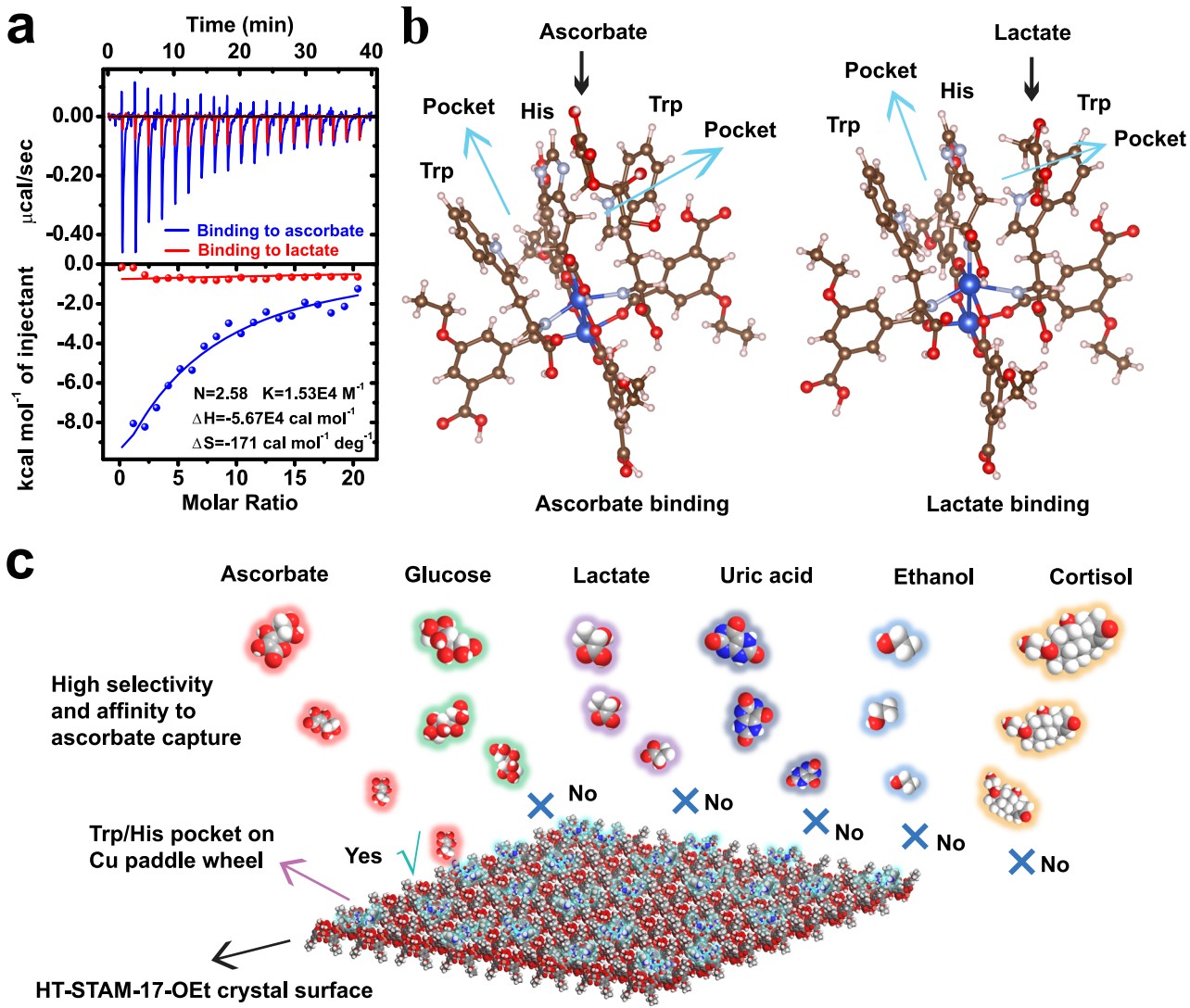

**Fig. 4 | Specific ascorbate recognition mechanism of HT-STAM-17-OEt. a** ITC curves of HT-STAM-17-OEt binding to ascorbate and lactate. **b** Optimized geometric structure for ascorbate and lactate adsorption. **c** The specific capture of sweat ascorbate by Trp/His on the STAM-17-OEt surface.

potential of 0.5 V. The −OH groups can be divided into two types: adsorbed water (3726 and 3703 cm$^{-1}$) and free $H_2O$ in MOF pores (3625 and 3599 cm$^{-1}$). Accordingly, the activation of HT-STAM-17-OEt is accompanied by the ionization of $H_2O$, which drives the formation of intermediates with strong oxidation. The collected HT-STAM-17-OEt powders after activation were characterized by XPS. The binding energies of Cu $2p_{3/2}$ and Cu $2p_{1/2}$ decrease by 0.5 and 0.4 eV, respectively (Fig. 5c), indicating different Cu coordination after electrical activation. This means that ionization-induced oxidation state generation is also accompanied by a partial change in the Cu paddle wheel structure.

Owing to the huge tendency of $H_2O$ adsorption on the oxygen of −C−O−Cu− in the Cu paddle wheel[46], $H_2O$ ionization occurs on this site. Fourier transform alternating current voltammetry (FT ac voltammetry), as a technique for studying the underlying electron transfer reaction[47,48], is employed to verify the structural variation. Harmonics at all levels from FT ac voltammetric data for HT-STAM-17-OEt electrocatalyst are displayed in Fig. 5d. The fundamental ac harmonic shows two well-defined processes for Cu(I/II) at 0.4/0.1 V and Cu(0/I) at −0.4/−0.1 V. The asymmetric shape of the harmonics demonstrates the coupling of a fast chemical reaction to electron transfer. A Cu paddle wheel structure with positive charges can be confirmed as a catalytic oxidation center based on the above

experimental results, and the positive charges stem from electrical field-induced ionization of $H_2O$ (Fig. 5e). Integrating these results with the XPS results, the involved chemical reaction should be the coordination of $H_2O$ with unsaturated Cu(II) generated from pulling the positively charged ligand. Through high-resolution mass spectrometry (HRMS) analysis, the oxidation product of ascorbate by HT-STAM-17-OEt was confirmed to be 2,3-diketo-L-gulonic acid (Supplementary Fig. 34). The whole ascorbate oxidation process can be summarized, as exhibited in Fig. 5f. Once ascorbate is captured by Trp/His on the Cu paddle wheel, a positively charged Cu paddle wheel with strong oxidability would push ascorbate to quickly dehydrogenate and generate dehydroascorbic acid. Then, the dehydroascorbic acid is further oxidized through a ring-opening reaction and produces 2,3-diketo-L-gulonic acid. After oxidation, the final product 2,3-diketo-L-gulonic acid would escape from the Trp/His binding sites, leaving space for the next ascorbate molecule. Despite the FT ac voltammetry results hinting that the Cu paddle wheel in HT-STAM-17-OEt after positive potential activation would not return to its initial structure under negative potential, the ascorbate electro-oxidation performance of HT-STAM-17-OEt can still be maintained over a long working time.

In addition to chemical structures, how the morphology of MOF crystals substantially influences the electrocatalytic performance also

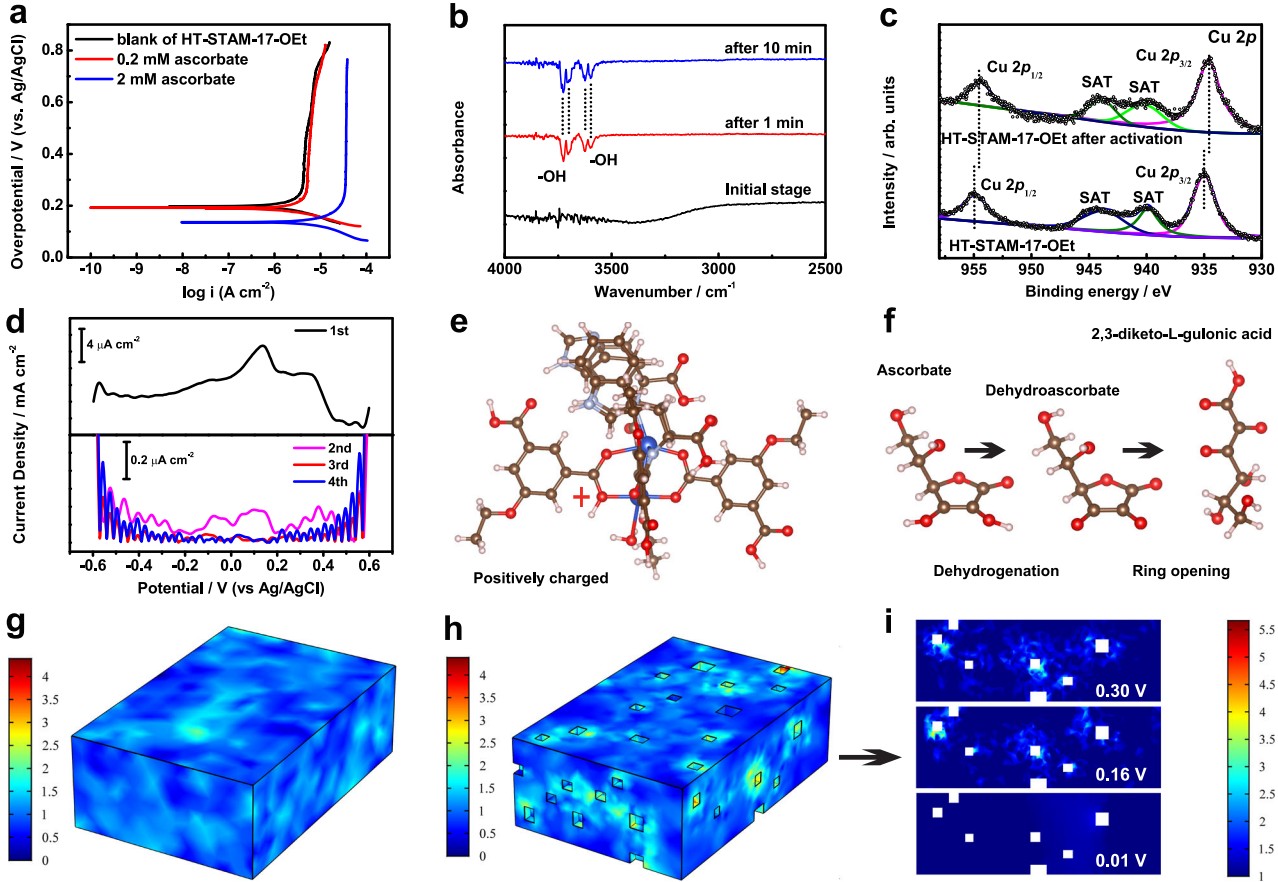

**Fig. 5 | Electrochemical ascorbate oxidation mechanism of HT-STAM-17-OEt.** **a** Potentiodynamic polarization curves of HT-STAM-17-OEt in 0.1 M NaCl (pH = 7) in the absence and presence of ascorbate. **b** In situ EC-IR spectrum of HT-STAM-17-OEt in 0.1 M NaCl (pH = 7) at different times at 0.5 V vs. Ag/AgCl. **c** Cu 2*p* XPS profiles of HT-STAM-17-OEt. d Resolved 1st, 2nd, 3rd, and 4th harmonic components of AC voltammograms for HT-STAM-17-OEt in 0.1 M NaCl (*f* = 9 Hz, Δ*E* = 0.080 V).

**e** Positively charged Cu paddle wheel as an oxidation center. **f** Electrooxidation process of ascorbate. The blue, red, brown, and gray balls represent Cu, O, C, and H atoms, respectively. Finite element simulated current distribution on the **g** 3D STAM−17-OEt microslate, **h** 3D HT-STAM-17-OEt with cubic defects, and **i** 2D HT-STAM-17-OEt layer with cubic defects at different overpotentials.

needs to be comprehended. Considering that the catalytic activity is governed by the electrical field, it is desirable to explore the relationship between the applied electrical field and Faraday current on the MOF geometry surface. To this end, two microscale geometry models of STAM-17-OEt and HT-STAM-17-OEt have been built as anodes to study the current density distribution under an electrical field by finite element simulation (see details in Supplementary Information). Because of limited active sites, we use the concentration-dependent Butler–Volmer equations[49]:

$$i = i_0 \left( C_{\mathrm{R}} e^{\frac{\alpha_a F \eta}{RT}} - C_{\mathrm{O}} e^{\frac{-\alpha_c F \eta}{RT}} \right) \qquad (1)$$

where $i_0$ is the exchange current density, $i$ is the charge transfer current density, $\eta$ is the overpotential, $R$ is the gas constant, $T$ is the temperature, and $C_{\mathrm{R}}$ and $C_{\mathrm{O}}$ are the reduction and oxidation species coefficients, respectively. $\alpha_a$ and $\alpha_c$ are the anode and cathode charge transfer coefficients, respectively. The dimensionless current density presents a completely different distribution pattern on the two geometric model surfaces at the same boundary conditions when the whole electrochemical system reaches a steady state. As seen in Fig. 5g, the dimensionless current density on the smooth cuboid surface shows a "lake-like" distribution at a potential of 0.5 V, and the distribution is relatively random. However, for the geometry of HT-STAM-17-OEt, the dimensionless current density on and around the cubic defect region is higher than that on the other

regions at a potential of 0.5 V (Fig. 5h). We have further discovered that the dimensionless current density around these cubic holes is dramatically enhanced with increasing overpotential (Fig. 5i). These simulation results illustrate that the introduction of cubic defects would directly change the surface Faraday current distribution on STAM-17-OEt. The current density around such geometric defects is highly "sensitive" to overpotential variation. This discovery also delivers an important point that the geometric defects of MOF crystals are the most catalytic active sites in the electrocatalytic process.

## Discussion
In summary, we customize an ascorbate oxidase-mimicking MOF HT-STAM-17-OEt for selective sweat ascorbate sensing. This MOF could specifically capture ascorbate from sweat biomolecules and electrooxidize, showing a high sensitivity of 0.18 and 0.48 mA cm$^{-2}$ mM$^{-1}$ in acidic and alkaline sweat, respectively. Tryptophan/histidine on a copper paddlewheel with high affinity selectively recognizes ascorbate by multiple hydrogen bonding interactions, determining the specific and targeted identification behavior. The electro-oxidation mechanism of HT-STAM-17-OEt is that the electrical field triggers the ionization of H$_2$O to generate a positively charged copper paddlewheel with a strong oxidation capability. This work provides MOF nanoenzymes with specific performance for sweat sensors, which may inspire significant understanding in mimicking natural enzymes for biological and healthcare applications.

## Methods

### Synthesis of Cu₂(OH)₃Cl nanocuboids, STAM-17-OEt and HT-STAM-17-OEt

The $Cu_2(OH)_3Cl$ nanocuboids were synthesized with a hydrolysis method. $CaCO_3$ flexible paper (2 cm × 3 cm) as the substrate was immersed in 0.5 M $CuCl_2$ aqueous solution for 24 h at room temperature. For this process, the light green $Cu_2(OH)_3Cl$ nanocuboids would grow on the surface of $CaCO_3$ paper. Then, the $CaCO_3$ flexible paper was removed and washed with water. After that, the substrate was immersed in ethanol to collect $Cu_2(OH)_3Cl$ nanocuboids by an ultrasonic method. Next, the collected $Cu_2(OH)_3Cl$ nanocuboids were washed, centrifuged, and dried under the air. The synthesis of STAM-17-OEt single crystals was performed by using $Cu_2(OH)_3Cl$ nanocuboids and 5-ethoxyisophthalic acid as the copper source and organic ligand, respectively. The conversion solution was prepared by adding 10 mg $Cu_2(OH)_3Cl$ nanocuboids and 19.684 mg 5-ethoxyisophthalic acid to 10 mL $H_2O$. This solution was held at 40 °C for 12 h without stirring. After the reaction, the blue STAM−17-OEt microslates were obtained by centrifugation and washing. For the postsynthesis treatment, the resultant 4 mg STAM−17-OEt microslates were dispersed in 20 mL $H_2O$ containing 20.4 mg tryptophan (Trp) and 7.75 mg histidine (His) for 12 h at room temperature. After washing and drying, the product HT-STAM−17-OEt with cubic defects was obtained.

### Sweat collection

Sweat samples were obtained from one healthy volunteer (male) who provided informed consent, and it is confirmed that they complied with all ethical regulations.

### Statistics and reproducibility

Each experiment's results are indicated in Fig. 2c–e, Supplementary Figs. 1a–d; 2; 3a–d; 5a, b; 6a-f; 7a, b was repeated three times independently with similar results.

### Reporting summary

Further information on research design is available in the Nature Portfolio Reporting Summary linked to this article.

## Data availability

The data that support the findings of this study are available from the corresponding author upon request.

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

## Acknowledgements

This work is supported by the National Key Research and Development Program of China (2021YFA1501000), the China Postdoctoral Science Foundation (2021M701292), the National Natural Science Foundation of China (52171069, 22005109), the Key Laboratory of Material Chemistry for Energy Conversion and Storage, Ministry of Education (2021), and the Open Project Program of Hubei Key Laboratory of Materials Chemistry and Service Failure (2020MCF02). We thank all the funding for supporting this work. We also acknowledge the measurement support of the Analytical and Testing Center of Huazhong University of Science and Technology (HUST), the Analytical and Testing Centre of the School of Chemistry and Chemical Engineering (HUST), and Research Core Facilities for Life Science (HUST).

## Author contributions

H.L. and B.Y.X. conceived and supervised the project. Z.W. designed and performed the experiments, including sample synthesis, characterization, and measurements. Z.W. performed finite element analysis with COMSOL Multiphysics (5.3a). K.X. performed DFT calculations with Materials Studio (8.0). Y.H., C.H., L.J., J.S., Z.R., J.Z., J.H., and F.X. performed the characterization and participated in the discussion. Z.W., H.L., and B.Y.X. co-wrote and revised the manuscript. All authors read and commented on this manuscript.

## Competing interests

The authors declare no competing interests.

## Ethics

We have complied with all relevant ethical regulations for human research participants. Ethics Committee of Clinical Drug Trials of Huazhong University of Science and Technology provided guidelines for study procedures.
