## [Peer Review File · Nature Communications]

Natural Oxidase-Mimicking Copper-Organic Frameworks for Targeted Identification of Ascorbate in Sensitive Sweat SensingReviewers' Comments:

Reviewer #1:

Remarks to the Author:

Electrochemical-based sweat sensors are capable of monitoring physiological information in real time and thus offer applicaiton potential for personalized healthcare. However, the existing natural oxidases and nanoenzymes suffer from instability and low selectivity, which largely limit the precise monitoring of targeted sweat biomolecules. In this work, the authors report histidine (His)- and tryptophan (Trp)-functionalized MOFs (HT-STAM-17-OEt) by epitaxial growth and post-synthetic treatments with amino acids, which show a high sensitivity to ascorbate molecules. This work provides a systematic and comprehensive electrochemical analysis, and is highly valuable as a reference for developing bionic MOF materials. However, there are a number of major issues regarding theoretical calculations, mechanism etc. which need to be carefully addressed.

1. Are the amino acids loaded on the surface or in the pores of STAM-17-OEt? How did the authors obtain the ratio information of histidine and tryptophan? The authors should characterize the loading sites and capacity, and also provide solid evidence for their claim.
2. There are many possible forms of loaded amino acids. Did the authors test other possible models? The authors should give specific reasons for choosing the current DFT model.
3. The information and references of functional, basis sets and other parameters (such as dispersion correction method, fixed atoms during optimization, etc.) of DFT calculations should be provided. Otherwise, it is meaningless to discuss the results. The adsorption energies of ascorbate and lactate are useful for elucidating the selectivity, which are expected to be calculated.
4. What are the products of electrooxidation? The authors need to add the composition of products and information of catalytic process.
5. In the selectivity test (Figure S15), the concentration of molecules are different, and the interference of glucose and cortisol tested at lower concentration is not clear. In addition, glucose and cortisol were tested but not mentioned in the manuscript, the discussion needs to be extended.
6. The full name of CP should be provided.

Reviewer #2:

Remarks to the Author:

The authors reported an ascorbate oxidase-mimicking MOF HT-STAM-17-OEt for selective sweat ascorbate sensing. MOF HT-STAM-17-OEt nanozyme was developed with a well structural bionic design of corresponding enzymes which is one of the strengths of the manuscript. The upgraded sweat sensors for bio-related detection are regarded as having a great application prospect in healthcare. Overall, this work can inspire more design ideas of nanozymes for sensing technologies and beyond. Therefore, I would like to recommend this work to publish. Below are a few suggestions for the authors.

1. "Among many sweat biomolecules, ascorbate is one of the most important biomarkers related to nutrition and immunity." This part should be explained more to tell the importance of this ascorbate sensor.
2. The background of nanozymes and ascorbate oxidase-like activity is not informative. A few studies have investigated the ascorbate oxidase-like activity of nanozyme (J. Mater. Chem. B 2016, 4, 7895-7901; ChemBioChem 2020, 21, 978-984; Sens. Actuators B: Chem. 2021, 347, 130627). However, compared with natural enzymes, these nanozymes lack substrate selectivity. Efforts to obtain ascorbate oxidase-like nanozyme with specificity are highly required. About this issue, designing nanozymes by simulating the structure of natural enzymes is surely a meaningful strategy (J. Am. Chem. Soc. 2020, 142, 15569-15574). These works related to this area might be helpful to the revision.
3. How about the product of the ascorbate oxidation catalyzed by natural ascorbate oxidase or MOF HT-STAM-17-OEt respectively? The catalytic process and mechanism should be further explained.
4. The affinity of ascorbate oxidase or MOF HT-STAM-17-OEt towards ascorbate should be compared.

5. The MOF HT-STAM-17-OEt-based ascorbate sensor exhibits good repeatability and stability. How about the stability of MOF HT-STAM-17-OEt compared with natural ascorbate oxidase?
6. Please take the cost of sweat sensor into consideration and add relevant results.
7. The inset figures in figure 3c are not clear enough. The equation in Page S-5 is unclear. The font in the scale of Figure S6 is small. Please adjust them.

Reviewer #3:

Remarks to the Author:

This study reported an amino acid functionalized MOF as a sensitive electrochemical sensor for ascorbate detection in sweat. Being able to accurately and timely detect tracing biomolecules from sweat is of great importance in the development of new wearable biomedical devices. This study designed a copper paddle wheel type MOF, and with Trp and His functionalization the MOF showed good detection specificity for ascorbate. However, with the current results, unfortunately I cannot recommend its acceptance in Nature Communications. With so many new developments of electrochemical and wearable sensors sweat, what's the novelty and advantage of the reported system? With the ability of sensing just one common biomolecule from the sweat, it is probably not enough for such a high impact journal like Nature Communications.

-The mechanism of Trp and His functionalization on the MOF is not clearly demonstrated.

-The authors claim that Trp and His on the MOF has specific binding to ascorbate. But without a strategy to control the spatial location of Trp and His on the MOF, it is unclear how the binding specificity is achieved.

-In addition, Trp and His binding site to ascorbate is distinct from active copper site, so how the overall mechanism from binding ascorbate to these amino acids then transfer the ascorbate to the copper center is not clear.

-It is claimed as ascorbate oxidase-mimicking, but no direct comparison of performance with natural enzyme is done.

-Human sweat is very complex, therefore more complex artificial solutions with a mixture of potential interfering molecules should be systematically studied.

Response to referees

Reviewer #1:

Electrochemical-based sweat sensors are capable of monitoring physiological information in real time and thus offer application potential for personalized healthcare. However, the existing natural oxidases and nanoenzymes suffer from instability and low selectivity, which largely limit the precise monitoring of targeted sweat biomolecules. In this work, the authors report histidine (His)- and tryptophan (Trp)-functionalized MOFs (HT-STAM-17-OEt) by epitaxial growth and post-synthetic treatments with amino acids, which show a high sensitivity to ascorbate molecules. This work provides a systematic and comprehensive electrochemical analysis, and is highly valuable as a reference for developing bionic MOF materials. However, there are a number of major issues regarding theoretical calculations, mechanism etc. which need to be carefully addressed.

1. Are the amino acids loaded on the surface or in the pores of STAM-17-OEt? How did the authors obtain the ratio information of histidine and tryptophan? The authors should characterize the loading sites and capacity, and also provide solid evidence for their claim.

Reply: Thank you for your question and constructive suggestions. No distinct XRD peak variation is observed in HT-STAM-17-OEt, indicating that the whole crystal structures of STAM-17-OEt are well retained after Trp/His treatment. This also means that the ligand exchange reaction does not occur in this case. Smooth and flat internal exfoliation layers observed by high-resolution transmission electron microscopy (HRTEM) (Figure 2b) and uniform N element mapping (Figure S6) have demonstrated that the amino acids tend to load on the surface of STAM-17-OEt, rather than internal crystals or the pores.

We further performed attenuated total reflectance infrared (ATR-IR) spectroscopy to characterize His, Trp, STAM-17-OEt, and HT-STAM-17-OEt, aiming to obtain more information about His and Trp on STAM-17-OEt. As shown in Figure 2f, the peaks at 3387 and 3334 cm^{-1} correspond to N-H scissoring, and

the peak at 3269 cm^{-1} corresponds to NH stretching vibrations. The peaks at 2907 , 1226 , 845 , and 581 cm^{-1} are attributed to O-H stretching (COOH), C-NH- stretching, N-H in-plane rocking and O-H out-of-plane deformation in Trp, respectively. The peaks 1150 and 822 cm^{-1} are assigned to C=N-C stretching and N-H in-plane rocking vibrations in His, respectively. Apart from the characteristic bands of His and Trp, HT-STAM-17-OEt shows a new characteristic peak of N-Cu- (696 cm^{-1}), which is attributed to Cu-N stretching vibrations. Additionally, Cu-O stretching increases from 490 to 494 cm^{-1} , which originates from the influence of the coordination of NH_2 to the Cu paddle wheel structure. The ATR-IR results demonstrate that His and Trp attack the Cu paddle wheel and coordinate to Cu on STAM-17-OEt. Therefore, the surficial Cu paddle wheel on STAM-17-OEt could be confirmed as the loading site for His and Trp.

According to the proportional relation between absorbance and concentration, the ratio of the infrared absorbance coefficient for C-NH- (stretching)/N-H (in-plane rocking vibration) in Trp and His can be calculated as 1.3 by normalization of the common peak (O-H out-of-plane bending) in Trp and His. The absorbance ratio of C-NH- (stretching)/N-H (in-plane rocking vibration) in HT-STAM-17-OEt was further calculated to be 2.44. Therefore, the ratio of Trp/His was calculated as 1.88, which is consistent with the ratio of amino acids (2:1) used for treatment.

We have added more content about amino acid modification in the revised manuscript. We have included the ATR-IR figure (STAM-17-OEt and HT-STAM-17-OEt) in the manuscript as a new Figure 2f. The previous Figure 2f (AFM-IR) is presented in the supporting information (Figure S9).

Figure 2f. ATR-IR spectra of STAM-17-OEt and HT-STAM-17-OEt.

Figure S8. ATR-IR spectra of His and Trp.

2. There are many possible forms of loaded amino acids. Did the authors test other possible models? The authors should give specific reasons for choosing the current DFT model.

Reply: Thank you for your reminder. For previous DFT calculations, the single Cu paddle wheel cluster was taken as the model for studying amino acid loading. In the previous model, the amino acids were connected to carbonyls because we considered the -NH-CO- groups, which originated from the reaction between Trp and exposed carboxyl groups. Based on ATR-IR results, we discover that the previous DFT model is not rigorous enough because there may be not enough carboxyl on the MOF surface for forming -NH-CO-. ATR-IR proves the -HN-Cu generation for HT-STAM-17-OEt, and its generation originates from two

aspects. One is direct axial coordination of amino acids. The other is that amino acids connect to the ligand through destruction on the surface Cu paddle wheel due to the Cu Lewis acid property-induced activation of carbonyl carbon. Consequently, a new more reasonable model has been proposed for DFT calculations to explain selectivity. Specific reasons for choosing the DFT model are also given.

We discovered that STAM-17-OEt treated only with Trp or His still responded to lactate (Figure S17). This means that the structures composed of the same amino acids His or Trp on the STAM-17-OEt surface still have high affinity for lactate. The structures composed of His and Trp on the STAM-17-OEt surface determine ascorbate specificity due to higher affinity to ascorbate. Then, the influence of the Trp/His modification sequence on selectivity was further explored. We have discovered STAM-17-OEt modified by first using Trp and then His still has lactate interference. STAM-17-OEt modified by first using His and then Trp also has lactate interference. This result indicates that His/Trp structures that can preferentially bind to ascorbate would not form by changing the modification order owing to full occupation of Trp or His on the Cu paddle wheel. Integrating with the ATR-IR results, there are six possible structures of amino acids on the Cu paddle wheel, as shown below (Figure S27). These structures have two open pockets consisting of Trp and His for binding two ascorbate molecules, consistent with the ITC result that the average number of ascorbate binding sites per Cu paddle wheel is 2.58. Among these six structures, structures of a, b, c and d with the same amino acid interaction are excluded due to lactate interference. The structures of e and f are reasonable because the central amino acid can be shared by two different amino acids, ensuring the combined effect of Trp/His. However, due to the overall ratio of Trp/His (2:1) on the STAM-17-OEt surface, structure f is more reasonable than structure e and dominates on the MOF surface due to its selectivity. By using Trp/His to modify other Cu-MOFs with Cu paddle wheel structures, ascorbate selectivity was found to be a general phenomenon (Figure S28), confirming the selective binding and catalytic role of the Cu paddle wheel with Trp/His. Based on the above, the Cu paddle wheel structure with Trp/His(structure f) as a cluster

structure was built for DFT calculations: one His was connected to Cu by Cu-N coordination, and two Trp were connected to the ligand by -N-C- on both sides for coordination to Cu. The geometrically optimized structure f is shown in Figure S29.

Figure S26. Selectivity test of STAM-17-OEt treated with His (2.5 mM) and Trp (5 mM) in different order.

The applied potential is 0.5 V (vs. Ag/AgCl).

Figure S27. Six possible structures formed by Trp and His on the surface Cu paddle wheel.

Figure S28. Selectivity test of (a) CuBDC and (b) CuNDC before and after treatment with mixed 5 mM Trp/2.5 mM His. The applied potential is 0.5 V (vs. Ag/AgCl). CuBDC is composed of benzene-1,4-dicarboxylic acid (BDC) and Cu ions. CuNDC is composed of 1,4-naphthalenedicarboxylic acid (NDC) and Cu ions.

Figure S29. Geometric optimized structure f.

3. The information and references of functional, basis sets and other parameters (such as dispersion correction method, fixed atoms during optimization, etc.) of DFT calculations should be provided. Otherwise, it is meaningless to discuss the results. The adsorption energies of ascorbate and lactate are useful for elucidating the selectivity, which are expected to be calculated.

Reply: Thank you for pointing out this. The information and references of functional, basis sets and other parameters for DFT calculations have been provided in the revised supporting information. First-principles calculations were performed within the framework of density functional theory (DFT). The linear combination of the atomic orbital method and generalized gradient approximation GGA/PW91 functional were employed. Double numerical plus polarisation (DNP) was employed as the basis set. The core electrons were obtained using the all electron method considering all electrons in the system. The DFT+D method within the OBS scheme was adopted owing to the van der Waals (vdW) weak interaction. The self-consistent field (SCF) tolerance was set as 1×10^{-5} Hartree. For geometric optimization, the DFT calculations were performed at a spin-unrestricted set. The amino acids and Cu paddle wheel atoms were free, and the ligand atoms were constrained.

By using structure f for geometric optimization, the adsorption energies of ascorbate and lactate were further calculated as -0.78 and 1.74 eV (Figure S3), respectively, suggesting that ascorbate capture on the Cu paddle wheel is a spontaneous process, whereas lactate capture requires additional energy. Calculation results show that ascorbate binding occurs more easily than lactate binding because of the more negative Gibbs free energy. A corresponding description has been added to the revised manuscript (Page 10).

4. What are the products of electro-oxidation? The authors need to add the composition of products and information of catalytic process.

Reply: Thank you for your attention to this issue. Following your suggestion, we further collected the electrolyte containing the product of ascorbate oxidation by HT-STAM-17-OEt MOF and performed high-resolution mass spectrometry (HRMS). The result shows a distinct peak at a mass/charge (m/z) of 215.0170, which corresponds to 2,3-diketo-L-gulonic acid. Thus, the oxidation product of ascorbate by HT-STAM-17-OEt was confirmed to be 2,3-diketo-L-gulonic acid. The catalytic process can be summarized as follows (Figure 5f): once ascorbate is captured by Trp/His on the Cu paddle wheel, a positively charged

Cu paddle wheel with strong oxidability would push ascorbate to quickly dehydrogenate and generate dehydroascorbic acid. Then, the dehydroascorbic acid is further oxidized through a ring-opening reaction and produces 2,3-diketo-L-gulonic acid (Figure S33). After oxidation, the final product 2,3-diketo-L-gulonic acid would escape from the Trp/His binding sites, leaving space for the next ascorbate molecule. A corresponding description of the product and catalytic process has been added to the revised manuscript (Page 12).

Figure S33. High resolution mass spectrometry (HRMS) of ascorbate oxidation product by HT-STAM-17-OEt.

Figure 5f. Electrooxidation process of ascorbate.

5. In the selectivity test (Figure S15), the concentrations of molecules were different, and the interference of glucose and cortisol tested at lower concentrations was not clear. In addition, glucose and cortisol were tested but not mentioned in the manuscript, the discussion needs to be extended.

Reply: Thank you for your suggestion. The molecular concentration used for the selectivity test is their respective highest concentration in sweat. Following your suggestion, we have further unified the concentration and added 50 nM (low) interference molecules, including glucose and cortisol, as well as lactate, uric acid, and ethanol, to study selectivity at lower concentrations (Figure S15 in the revised supporting information). The results show that interfering molecules with lower concentrations also cannot generate any interference current signals for HT-STAM-17-OEt and STAM-17-OEt. Additionally, we have added discussions about glucose and cortisol in the revised manuscript, as seen on page 7: “these interfering biomolecules at low concentrations (50 nM) cannot generate any interference current signals for HT-STAM-17-OEt and STAM-17-OEt (Figure S15). Moreover, HT-STAM-17-OEt can quickly respond to ascorbate when ascorbate is added with high concentrations of interfering biomolecules at the same time, while STAM-17-OEt has no response in this case (Figure S16). This result demonstrates that HT-STAM-17-OEt can favorably recognize ascorbate from other biomolecules, but STAM-17-OEt cannot.

Figure S15. (a-b) Amperometry response of STAM-17-OEt and HT-STAM-17-OEt to 10 μM ascorbate with high low interference (50 nM) in 0.1 M NaCl (pH=7) at 0.5 V vs. Ag/AgCl.

6. The full name of CP should be provided.

Reply: Thank you for pointing out this. The CP is a copper paddle wheel, and we have used full name to replace abbreviation in the revised manuscript.

Reviewer #2:

The authors reported an ascorbate oxidase-mimicking MOF HT-STAM-17-OEt for selective sweat ascorbate sensing. MOF HT-STAM-17-OEt nanozyme was developed with a well structural bionic design of corresponding enzymes which is one of the strengths of the manuscript. The upgraded sweat sensors for bio-related detection are regarded as having a great application prospect in healthcare. Overall, this work can inspire more design ideas of nanozymes for sensing technologies and beyond. Therefore, I would like to recommend this work to publish. Below are a few suggestions for the authors.

1. “Among many sweat biomolecules, ascorbate is one of the most important biomarkers related to nutrition and immunity.” This part should be explained more to tell the importance of this ascorbate sensor.

Reply: Thank you for pointing out this. We have added more related content in the revised manuscript to emphasize the importance of the sweat ascorbate sensor: “Ascorbate plays an indispensable role in many body functions, such as iron absorption, collagen production, infection protection and neurological disorder prevention. Thus, ascorbate sensing in sweat is of significance in evaluating and preventing the risks of these diseases”, as shown in the revised manuscript (Page 4).

2. The background of nanoenzymes and ascorbate oxidase-like activity is not informative. A few studies have investigated the ascorbate oxidase-like activity of nanozyme (J. Mater. Chem. B 2016, 4, 7895-7901; ChemBioChem 2020, 21, 978-984; Sens. Actuators B: Chem. 2021, 347, 130627). However, compared with natural enzymes, these nanozymes lack substrate selectivity. Efforts to obtain ascorbate oxidase-like nanozymes with specificity are highly needed. Regarding this issue, designing nanozymes by simulating the structure of natural enzymes is surely a meaningful strategy (J. Am. Chem. Soc. 2020, 142, 15569-15574). These works related to this area might be helpful to the revision.

Reply: Thank you for pointing out this and providing these important references. We have added some content to the revised manuscript to make the background of nanoenzymes and ascorbate oxidase-like

activity more informative, as displayed on page 4. Although some nanoenzyme-related research works have demonstrated that CuO nanoparticles and Pt nanoparticles both have ascorbate oxidase-like activity and can efficiently catalyze ascorbate, these artificial nanoenzymes cannot specifically recognize ascorbate owing to chemical structure limitations. The corresponding references have also been cited (ref 24, 27-30, 33-35).

3. How about the product of ascorbate oxidation catalyzed by natural ascorbate oxidase or MOF HT-STAM-17-OEt? The catalytic process and mechanism should be further explained.

Reply: Thank you for your question. Classic research works have confirmed that ascorbate can be oxidized by natural ascorbate oxidase to dehydrate and generate dehydroascorbate.^[1,2] Following your suggestion, we further collected the electrolyte containing the oxidation product ascorbate by HT-STAM-17-OEt and performed high-resolution mass spectrometry (HRMS). The result shows a distinct peak at a mass/charge (m/z) of 215.0170, which corresponds to 2,3-diketo-L-gulonic acid. Thus, the oxidation product of ascorbate by HT-STAM-17-OEt is confirmed to be 2,3-diketo-L-gulonic acid. The catalytic process can be summarized as follows (Figure 5f): Once ascorbate is captured by Trp/His on the Cu paddle wheel, a positively charged Cu paddle wheel with strong oxidability would push ascorbate to quickly dehydrogenate and generate dehydroascorbic acid. Then, the dehydroascorbic acid is further oxidized through a ring-opening reaction and produces 2,3-diketo-L-gulonic acid. After oxidation, the final product 2,3-diketo-L-gulonic acid would escape from the Trp/His binding sites, leaving space for the next ascorbate molecule. A corresponding description of the product and catalytic process has been added to the revised manuscript (Page 12).

Figure S33. High resolution mass spectrometry (HRMS) of ascorbate oxidation product by HT-STAM-17-OEt.

Figure 5f. Electrooxidation process of ascorbate.

4. The affinity of ascorbate oxidase or MOF HT-STAM-17-OEt towards ascorbate should be compared.

Reply: Thank you for your suggestion. The affinity of ascorbate oxidase towards ascorbate was also further evaluated by isothermal titration calorimetry (ITC). According to the ITC results, four different affinities of ascorbate oxidase to ascorbate can be achieved because the surrounding microenvironment of these four binding sites in ascorbate oxidase is different. The affinity of ascorbate oxidase to ascorbate is 5.98×10^4 , 1.03×10^5 , 1.88×10^3 and 5.92×10^3 M^{-1} , respectively. Overall, the affinity of ascorbate oxidase is higher than that of HT-STAM-17-OEt (1.53×10^4 M^{-1}). We have moved the ITC figure of ascorbate oxidase to ascorbate in the revised supporting information (Figure S25b).

Figure S25b. ITC curve of ascorbate oxidase binding to ascorbate.

Part of Figure 4a. ITC curve of HT-STAM-17-OEt binding to ascorbate.

5. The MOF HT-STAM-17-OEt-based ascorbate sensor exhibits good repeatability and stability. How about the stability of MOF HT-STAM-17-OEt compared with natural ascorbate oxidase?

Reply: Thank you for your attention to this issue. Following your suggestion, we have further prepared a natural ascorbate oxidase (AOx)-based electrode, and the AOx electrode shows poor performance stability. Ascorbate oxidase catalyzes the oxidation of ascorbate to dehydroascorbic acid by consuming oxygen. Hence, for the AOx-based electrode, quantification of the increasing ascorbate concentration is performed by correlating the decreased oxygen-reduction current. As shown in the CV curve below (Figure S22a), the reduction current of oxygen decreases in the presence of ascorbate (-0.5 to -0.8 V). We further used a chronoamperometric test to evaluate the static current response of the AOx electrode. Under the condition of oxygen saturation, the AOx electrode shows a high sensitivity ($2.0 \text{ mA cm}^{-2} \text{ mM}^{-1}$) at the initial stage. However, the reduction current did not further decrease after three additions of $10 \text{ }\mu\text{M}$ ascorbate, demonstrating ascorbate oxidases had already lost activity (Figure S22b). Because ascorbate oxidases, as enzymes, are highly sensitive to the external environment, including temperature and pH, an inappropriate external environment would result in irreversible structural changes in ascorbate oxidases, which is the origin of inactivation.

In comparison, the sensitivity of MOF HT-STAM-17-OEt is stable from low to high concentrations. There was no sensitivity degradation after 70 catalytic cycles for HT-STAM-17-OEt (Figure S21). Owing to its more stable coordination structures, HT-STAM-17-OEt can resist changes in the external environment and exhibit much higher stability than natural ascorbate oxidases. We also put the performance figures for the AOx electrode in the revised supporting information (Figure S22).

Figure S22. (a) CV curves at 20 mV s⁻¹ and (b) amperometry response at -0.6 V vs. Ag/AgCl for the AOx electrode in 0.1 M NaCl (pH=7).

6. Please take the cost of sweat sensor into consideration and add relevant results.

Reply: Thank you for your question. We have added the cost of sweat sensors to the revised supporting information (page S3). The cost of the sweat sensor was evaluated according to the major electrode material cost. The costs of the MOF ligand, AgCl reference and Pt counter electrode are 1300, 350, and 1200 RMB/g, respectively. The masses of ligand, AgCl, and Pt for the sweat sensor are 0.6 mg, 2 mg and 5 mg, respectively. Thus, the cost of a single sweat sensor is 7.5 RMB (0.8 RMB for HT-STAM-17-OEt, 0.7 RMB for Ag/AgCl, 6 RMB for Pt). The Pt electrode accounts for most of the sensor cost. This means that developing stable and cheaper counter electrode materials is essential to cost reduction and commercialization of sweat sensors in the future.

7. The inset figures in figure 3c are not clear enough. The equation in Page S-5 is unclear. The font in the scale of Figure S6 is small. Please adjust them.

Reply: Thank you for carefully checking. We have modified the inset figures in Figure 3C, the equation, and the font in the scale of Figure S6 to make them clearer in the revised manuscript and supporting information.

Reviewer #3:

This study reported an amino acid-functionalized MOF as a sensitive electrochemical sensor for ascorbate detection in sweat. Being able to accurately and timely detect tracing biomolecules from sweat is of great importance in the development of new wearable biomedical devices. This study designed a copper paddle wheel-type MOF, and with Trp and His functionalization, the MOF showed good detection specificity for ascorbate. However, with the current results, unfortunately I cannot recommend its acceptance in Nature Communications. With so many new developments of electrochemical and wearable sensors, what is the novelty and advantage of the reported system? With the ability to sense just one common biomolecule from sweat, it is probably not enough for such a high impact journal as Nature Communications.

Reply: We sincerely thank the relatively critical comments and questions. Although various electrochemical sweat sensors have been reported, the deficiencies of sensing materials for target biomolecules have been completely overlooked in the current study. The sensing of sweat biomolecules is dependent on the recognition and oxidation behavior of corresponding oxidases. However, owing to their environmentally sensitive activities, oxidases lack stability and are rapidly inactivated, resulting in sweat sensor failure. Consequently, customizing nanoenzymes that can specifically respond to biomarkers is essential and challenging for sweat biomolecule sensors since common nanoenzymes have no specificity to target biomolecules. In this study, we customize the MOF nanoenzymes by mimicking the natural oxidase structure to achieve specific capture and electro-oxidation of sweat ascorbate, which is an important biomarker related to nutrition and immunity. By histidine (His)- and tryptophan (Trp)-functionalization, the MOF HT-STAM-17-OEt shows highly specific activity for ascorbate oxidation and superior stability than ascorbate oxidases, which can serve as an advanced sensing element in sweat ascorbate sensors. Furthermore, through supplemented experiments, customizing ascorbate nanoenzymes by using His and Trp functionalization to treat Cu-MOFs has been demonstrated as a general strategy to realize selectivity to

ascorbate. This work demonstrates a significant MOF nanoenzyme paradigm inspired by natural oxidase, which also provides an understanding for targeted identification in sweat sensing technologies.

1. The mechanism of Trp and His functionalization on the MOF is not clearly demonstrated.

Reply: Thank you for your question. No distinct XRD peak variation is observed in HT-STAM-17-OEt, indicating that the whole crystal structures of STAM-17-OEt are well retained after Trp/His treatment. This also means that the ligand exchange reaction does not occur in this case. Smooth and flat internal exfoliation layers observed by high-resolution transmission electron microscopy (HRTEM) (Figure 2b) and uniform N element mapping (Figure S6) have demonstrated that the amino acids tend to load on the surface of STAM-17-OEt, rather than internal crystals or the pores. We further performed attenuated total reflectance infrared (ATR-IR) spectroscopy to characterize STAM-17-OEt and HT-STAM-17-OEt to clearly study the functionalization mechanism, as shown below. Apart from the characteristic bands of His and Trp, HT-STAM-17-OEt shows a new characteristic peak of N-Cu- (696 cm^{-1}), which is attributed to Cu-N stretching vibrations. Additionally, Cu-O stretching increases from 490 to 494 cm^{-1} , which originates from the influence of the coordination of NH_2 to the Cu paddle wheel structure. The ATR-IR results demonstrate that His and Trp attack the Cu paddle wheel and coordinate to Cu on STAM-17-OEt. Therefore, the surficial Cu paddle wheel on STAM-17-OEt could be confirmed as the loading site for His and Trp. We have added a corresponding description in the revised manuscript and replaced AFM-IR with ATR-IR results in Figure 2f.

Figure 2f. ATR-IR spectra of STAM-17-OEt and HT-STAM-17-OEt.

2. The authors claim that Trp and His on the MOF have specific binding to ascorbate. However, without a strategy to control the spatial location of Trp and His on the MOF, it is unclear how the binding specificity is achieved.

Reply: Thank you for your question. Following your suggestion, we have further performed many experiments to study the origin of ascorbate binding specificity and the selectivity controlling strategy. First, we compared the binding behavior of HT-STAM-17-OEt to ascorbate and lactate at the same concentration. As shown below, isothermal titration calorimetry (ITC) results show a highly low heat compensation for binding to lactate of HT-STAM-17-OEt, indicating that lactate is difficult to capture by HT-STAM-17-OEt due to Trp/His effects.

Figure 4a. ITC curve of HT-STAM-17-OEt binding to ascorbate and lactate.

Furthermore, we discovered that STAM-17-OEt treated only with Trp or His still responded to lactate (Figure S17). This means that the structures composed of His and Trp on the STAM-17-OEt surface, rather

than structures composed of the same amino acids, determine ascorbate specificity due to the higher affinity to ascorbate. ATR-IR results indicate that His and Trp could interact with the catalytic center Cu paddle wheel to form a new structure. Then, the influence of the Trp/His modification sequence on selectivity was further explored. STAM-17-OEt is modified by first using Trp, and then His still has lactate interference. STAM-17-OEt modified by first using His and then Trp also has lactate interference (Figure S26). This result indicates that His/Trp structures that can preferentially bind to ascorbate would not form by changing the modification order owing to full occupation of Trp or His on the Cu paddle wheel. Integrating with the ATR-IR results, there are six possible structures of amino acids on the Cu paddle wheel, as shown below. These structures have two open pockets consisting of Trp and His for binding two ascorbate molecules, consistent with the ITC result that the average number of ascorbate binding sites per Cu paddle wheel is 2.58. Among these six structures, structures of a, b, c and d with the same amino acid interaction are excluded due to lactate interference. The structures of e and f are reasonable because the central amino acid can be shared by two different amino acids, ensuring the combined effect of Trp/His. However, considering the overall ratio of Trp/His (2:1) on the STAM-17-OEt surface, structure f is more reasonable than structure e and dominates on the MOF surface due to its selectivity. By using Trp/His to modify other Cu-MOFs with Cu paddle wheel structures, ascorbate selectivity was found to be a general phenomenon (Figure S28), confirming the selective binding and catalytic role of the Cu paddle wheel with Trp/His.

On the other hand, ascorbate specificity can be successfully controlled by using different concentrations of Trp/His to treat STAM-17-OEt. When STAM-17-OEt is treated with low concentrations of Trp and His (0.5 mM for Trp and 0.25 mM for His), HT-STAM-17-OEt has no specificity to ascorbate, and lactate interference is still significant, which is due to the small amount of amino acids on the surface of STAM-17-OEt. When STAM-17-OEt is treated with high concentrations of Trp and His (50 mM for Trp and 25 mM for His), HT-STAM-17-OEt has no electro-oxidation activity to ascorbate or lactate (Figure

S18). This phenomenon is due to many more His and Trp molecules fully covering the formed structure that is responsible for selectivity.

Figure S26. Selectivity test of STAM-17-OEt treated with His (2.5 mM) and Trp (5 mM) in different order.

The applied potential is 0.5 V (vs. Ag/AgCl).

Figure S27. Six possible structures formed by Trp and His on the surface Cu paddle wheel.

Figure S28. Selectivity test of (a) CuBDC and (b) CuNDC before and after treatment with mixed 5 mM Trp/2.5 mM His. The applied potential is 0.5 V (vs. Ag/AgCl). CuBDC is composed of benzene-1,4-dicarboxylic acid (BDC) and Cu ions. CuNDC is composed of 1,4-naphthalenedicarboxylic acid (NDC) and Cu ions.

3. In addition, the Trp and His binding site to ascorbate is distinct from the active copper site, so how the overall mechanism from binding ascorbate to these amino acids and then transferring ascorbate to the copper center is not clear.

Reply: Thank you for your question. Surface Trp and His do not act as a filter screen to let ascorbic acid pass through and transfer to the copper center. Actually, ascorbate after binding by amino acids would be directly electrooxidized because the copper paddle wheel with Trp/His is a whole structure with binding and oxidation capability. The oxidation process can be summarized as follows: Once ascorbate is captured by Trp/His on the Cu paddle wheel, a positively charged Cu paddle wheel with strong oxidability would push ascorbate to quickly dehydrogenate and generate dehydroascorbic acid. Then, the dehydroascorbic acid is further oxidized through a ring-opening reaction and produces 2,3-diketo-L-gulonic acid. After oxidation, the final product 2,3-diketo-L-gulonic acid would escape from the Trp/His binding sites, leaving space for the next ascorbate molecule. The corresponding oxidation mechanism description has been added in the revised manuscript (Page 12).

4. It is claimed as ascorbate oxidase-mimicking, but no direct comparison of performance with natural

enzyme is done.

Reply: Thank you for pointing out this. Following your suggestion, we have further prepared a natural ascorbate oxidase (AOx)-based electrode and directly compared the performance differences. Ascorbate oxidase catalyzes the oxidation of ascorbate to dehydroascorbic acid by consuming oxygen. Hence, for the AOx-based electrode, quantification of the increasing ascorbate concentration is performed by correlating the decreased oxygen-reduction current. As shown in the CV curve below (Figure S22a), the reduction current of oxygen decreases in the presence of ascorbate (-0.5 to -0.8 V). We further used a chronoamperometric test to evaluate the static current response of the AOx electrode. Under the condition of oxygen saturation, the AOx electrode shows a high sensitivity ($2.0 \text{ mA cm}^{-2} \text{ mM}^{-1}$) at the initial stage. However, the reduction current did not further decrease after three additions of $10 \mu\text{M}$ ascorbate, demonstrating ascorbate oxidases had already lost activity (Figure S22b). Because ascorbate oxidases, as enzymes, are highly sensitive to the external environment, including temperature and pH, an inappropriate external environment would result in irreversible structural changes in ascorbate oxidases, which is the origin of inactivation.

Figure S22. (a) CV curves at 20 mV s^{-1} and (b) amperometry response at $-0.6 \text{ V vs. Ag/AgCl}$ for the AOx electrode in 0.1 M NaCl ($\text{pH}=7$).

In comparison, the sensitivity of MOF HT-STAM-17-OEt is stable from low to high concentrations. There was no sensitivity degradation after 70 catalytic cycles for HT-STAM-17-OEt (Figure S21). Owing to

its more stable coordination structures, HT-STAM-17-OEt can resist changes in the external environment and exhibit much higher stability than natural ascorbate oxidases. We also put the performance figures for the AOx electrode in the revised supporting information (Figure S22)

5. Human sweat is very complex; therefore, more complex artificial solutions with a mixture of potential interfering molecules should be systematically studied.

Reply: Thank you for your constructive suggestion. In our previous study, the concentrations of interfering molecules, including lactate, glucose, uric acid, ethanol, and cortisol, used for the selectivity study were their highest concentrations in human sweat. Following your suggestion, we have further systematically studied the static current response to ascorbate of HT-STAM-17-OEt in artificial salt solutions with different pH values (5-8) and mixtures of low (50 nM) and high (20 mM) concentrations of interfering molecules, as shown below. With the presence of low (50 nM) concentrated interfering biomolecules in salt solutions (Na^+ , K^+ , NH_4^+ , Ca^{2+} , Cl^-), HT-STAM-17-OEt shows ascorbate sensitivity of $0.18 \text{ mA cm}^{-2} \text{ mM}^{-1}$ in acidic solution and $0.48 \text{ mA cm}^{-2} \text{ mM}^{-1}$ in neutral (alkaline) solution. With the presence of high (20 mM) concentrated interfering biomolecules in salt solutions (Na^+ , K^+ , NH_4^+ , Ca^{2+} , Cl^-), the sensitivity remains at the same level in acidic solution ($0.18 \text{ mA cm}^{-2} \text{ mM}^{-1}$) and neutral (alkaline) solution ($0.48 \text{ mA cm}^{-2} \text{ mM}^{-1}$). These results demonstrate that a mixture of sweat biomolecules with high and low concentrations, including lactate, uric acid, glucose, ethanol, and cortisol, would not influence the response to ascorbate. We have also put the figures in the revised supporting information (Figure S24).

Figure S24. Amperometry response of HT-STAM-17-OEt to 50 μM ascorbate at 0.5 V vs. Ag/AgCl in different acid and basic salt solutions (0.1 M NaCl, 20 mM KCl CaCl₂ and NH₄Cl) containing (a) 50 nM lactate, uric acid, glucose, ethanol and cortisol and (b) 20 mM lactate, uric acid, glucose, ethanol and cortisol.

References

- [1] K. Tokuyama, C. R. Dawson, *Biochimica et Biophysica Acta* 1962, 56, 427-439.
- [2] H. G. Steinman, C. R. Dawson, *Journal of the American Chemical Society* 1942, 64, 1212-1219.

Reviewers' Comments:

Reviewer #1:

Remarks to the Author:

The authors have well addressed my questions. I only have two more comments: The infrared spectroscopy is considered as a qualitative characterization technique, so I don't feel the IR data are very convincing. There are many reliable and quantitative techniques, such as NMR, LC-MS etc., which are more suitable to characterize the quantity of loading amounts and ratios. In addition, the current evidence is not able to exclude the possibility of a dominant surface absorption in the pores, which needs to be further discussed and characterized.

Reviewer #2:

Remarks to the Author:

The authors have revised the manuscript according to the reviewer's comments. It is satisfied now to be published.

Reviewer #3:

Remarks to the Author:

Good job in addressing most of my and other reviewers comments. The quality is very much improved with the newly supplemented results.

Response Letter

Reviewer #1:

The authors have well addressed my questions. I only have two more comments: The infrared spectroscopy is considered as a qualitative characterization technique, so I don't feel the IR data are very convincing. There are many reliable and quantitative techniques, such as NMR, LC-MS etc., which are more suitable to characterize the quantity of loading amounts and ratios. In addition, the current evidence is not able to exclude the possibility of a dominant surface absorption in the pores, which needs to be further discussed and characterized.

Reply: Thanks for your good suggestions. Following your suggestion, the quantity loading amounts and ratios of amino acids are investigated by liquid chromatograph mass spectrometer (LC-MS). By dissolving the HT-STAM-17-OEt and removing Cu ions and ligands, the remaining Trp and His are tested (Figure S10). According to the established standard curve and dilution relationship (20 times), the loading amount of Trp and His on HT-STAM-17-OEt (100 mg) is calculated as 3.4776×10^{-2} and 1.7796×10^{-2} mmol, respectively. Thus, the mass loading of Trp and His is calculated as 7.10% and 2.76%, respectively. And the mole ratio of Trp/His can be counted as 1.95, which is consistent with ATR-IR result (Figure S8) and the ratio of amino acids (2:1) initially used for treatment. We put the LC-MS figures in revised supplementary information (Figure S10) and corresponding descriptions are added in the revised manuscript (Page 6).

Figure S10. (a) Liquid chromatograph for Trp and His with standard concentrations and (b-c) corresponding standard curves. The retention time of His and Trp is 3.05 and 5.16 min, respectively. (d) Liquid chromatograph of the remaining Trp and His with 20 times of dilution by dissolving STAM-17-OEt and removing other constituents. Mass spectrogram of the separated (e) Trp and (f) His.

As for possible surface absorption in the pores, theoretically, Trp or His molecules may diffuse into the micropores of MOFs and occupy the channels, but the amounts are small. The absorption of amino acid is easily to occur on the crystal surface because of its higher accessibility. If a large number of Trp and His can freely enter into the pores of STAM-17-OEt and interact with pore walls, the whole crystal structure would change significantly due to the strong etching effect of Trp/His. However, from current experimental results, we found no obvious changes in the MOFs crystal structures, as indicated by XRD patterns (Figure 2b) and TEM images (Figure 2e). Moreover, according to the TEM and elementary N mapping, the modification of amino acids is also realized only on the crystal surface (Figure S6). Moreover, due to the technology limitation, we didn't find some effective probe technologies which can precisely detect the amino acids. In summary, the above results suggest that amino acids are mainly adsorbed on the surface rather than the inner pores or channels. We also provide the additional results into revised supplementary information (Figure S10), and add the corresponding discussions in the revised manuscript (marked part, page 6).

Thank you very much!

Reviewer #2:

The authors have revised the manuscript according to the reviewer's comments. It is satisfied now to be published.

Reply: We are grateful for the reviewer #2 for his constructive comments and suggestions to improve the quality of this work. Thank you very much.

Reviewer #3:

Good job in addressing most of my and other reviewers comments. The quality is very much improved with the newly supplemented results.

Reply: The authors thank the Reviewer #3 for the positive comments and suggestions during the revision process. These comments really help us to improve our work. Thank you very much.

Reviewers' Comments:

Reviewer #1:

Remarks to the Author:

The authors have addressed all my questions, therefore I recommend the publication of the revision as it is.

Response Letter

Reviewer #1:

The authors have addressed all my questions, therefore I recommend the publication of the revision as it is.

Reply: Thanks for your kind review.